# Lack of sexual dimorphism in a mouse model of isoproterenol-induced cardiac dysfunction

**Marianne K. O. Grant[1], Ibrahim Y. Abdelgawad[1], Christine A. Lewis[1], Davis Seelig[2], Beshay N. Zordoky[1]***

1 Department of Experimental and Clinical Pharmacology, College of Pharmacy, University of Minnesota, Minneapolis, Minnesota, United States of America, 2 Department of Veterinary Clinical Sciences, College of Veterinary Medicine, University of Minnesota, St. Paul, Minnesota, United States of America

* zordo001@umn.edu

**Data Availability Statement:** All relevant data are within the paper and its Supporting Information files.

## Abstract

Sex-related differences in cardiovascular diseases are highly complex in humans and model-dependent in experimental laboratory animals. The objective of this work was to comprehensively investigate key sex differences in the response to acute and prolonged adrenergic stimulation in C57Bl/6NCrl mice. Cardiac function was assessed by trans-thoracic echocardiography before and after acute adrenergic stimulation (a single sub-cutaneous dose of isoproterenol 10 mg/kg) in 15 weeks old male and female C57Bl/6NCrl mice. Thereafter, prolonged adrenergic stimulation was achieved by sub-cutaneous injections of isoproterenol 10 mg/kg/day for 14 days in male and female mice. Cardiac function and morphometry were assessed by trans-thoracic echocardiography on the 15th day. Thereafter, the mice were euthanized, and the hearts were collected. Histopathological analysis of myocardial tissue was performed after staining with hematoxylin & eosin, Masson's trichrome and MAC-2 antibody. Gene expression of remodeling and fibrotic markers was assessed by real-time PCR. Cardiac function and morphometry were also measured before and after isoproterenol 10 mg/kg/day for 14 days in groups of gonadectomized male and female mice and sham-operated controls. In the current work, there were no statistically significant differences in the positive inotropic and chronotropic effects of isoproterenol between male and female C57Bl/6NCrl. After prolonged adrenergic stimulation, there was similar degree of cardiac dysfunction, cardiac hypertrophy, and myocardial fibrosis in male and female mice. Similarly, prolonged isoproterenol administration induced hypertrophic and fibrotic genes in hearts of male and female mice to the same extent. Intriguingly, gonadectomy of male and female mice did not have a significant impact on isoproterenol-induced cardiac dysfunction as compared to sham-operated animals. The current work demonstrated lack of significant sex-related differences in isoproterenol-induced cardiac hypertrophy, dysfunction, and fibrosis in C57Bl/6NCrl mice. This study suggests that female sex may not be sufficient to protect the heart in this model of isoproterenol-induced cardiac dysfunction and underscores the notion that sexual dimorphism in cardiovascular diseases is highly model-dependent.

**Funding:** Research reported in this publication was supported by the National Institutes of Health's National Center for Advancing Translational Sciences, grant UL1TR002494. The content is solely the responsibility of the authors and does not necessarily represent the official views of the National Institutes of Health's National Center for Advancing Translational Sciences.

**Competing interests:** The authors declare that they have no competing interests.

## Introduction

Cardiovascular diseases remain the leading cause of mortality in both men and women globally [1], despite the conventional dogma that women are more protected against cardiovascular diseases than men. Indeed, the incidence of cardiovascular diseases in post-menopausal women is equal to those in age-matched men [2]. Furthermore, there are a number of cardiovascular diseases that are more prevalent in women than in men, including Takotsubo cardiomyopathy [3, 4] and microvascular angina [5]. Importantly, there are marked sex differences in the pathogenesis and/or clinical presentation of several cardiovascular diseases [6]. For instance, despite similar overall risk of heart failure between men and women, heart failure with reduced ejection fraction (HFrEF) is more prevalent in men, while heart failure with preserved ejection fraction (HFpEF) is more prevalent in women [6, 7]. Although the rationale for the higher prevalence of these specific conditions in women is still not fully understood, coronary microvascular dysfunction and endothelial inflammation have been suggested to play key roles [6].

The conventional view is that sex-related differences in cardiovascular diseases are mainly attributed to male and female sex hormones [2]. Traditionally, estrogen has been thought to provide a protective effect, whereas testosterone has a detrimental cardiovascular effect [2]. Although this traditional view has offered the biological underpinning for several cardiovascular pathologies that are more prevalent and more severe in males [8, 9], it could not explain other clinical and preclinical observations including: the worse outcome of post-menopausal women receiving supplemental estrogen therapy which was mainly attributed to increased rate of thromboembolic events [10], in addition to post-ischemic cardioprotection in aromatase knock-out mice [11] and worsening of heart failure in castrated male experimental animals [12, 13] which suggest a cardioprotective role of androgens. In addition to sex hormones, recent studies suggest that inherent differences in sex chromosome genes may contribute to sex differences in cardiovascular diseases [14]. These studies, among others, strongly suggest that sexual dimorphism in cardiovascular diseases is highly complex in humans (reviewed in [6, 8, 14–16]) and thus warrants further investigations in a variety of models in laboratory animals.

Isoproterenol is non-selective beta-adrenergic agonist commonly used as a pharmacological agent to induce a spectrum of cardiac pathologies in laboratory animal models. A number of studies determined the sex-related differences in the phenotypic manifestations and molecular determinants in models of isoproterenol-induced cardiac hypertrophy with discrepant results. While some of these studies reported that female experimental animals were protected against cardiac hypertrophy and fibrosis in response to chronic administration of low isoproterenol doses (0.04 mg/kg/day for 6 months) [17, 18], other studies reported no difference in cardiac hypertrophy in response to a higher dose (30 mg/kg/day for 7 days) [19], and one study showed that female rats were more sensitive than males to cardiac fibrosis induced by moderate doses of isoproterenol (7.5 mg/kg/day for 3 weeks) [20]. These studies suggest that sex-related differences in the response to isoproterenol are dose-dependent. Since most of the previous work has focused on determining sex-related differences in isoproterenol-induced cardiac hypertrophy and fibrosis [17–20], in the current work, we comprehensively investigated key echocardiographic, molecular, and histopathologic sex differences in a model of cardiac dysfunction produced by prolonged administration of moderate doses of isoproterenol (10 mg/kg/day for 14 days) in male and female C57Bl/6NCrl mice.

## Methods

### Animals

Animal procedures were approved by the Institutional Animal Care and Use Committee at the University of Minnesota (Protocol ID: 1807-36187A). Intact male ($n = 19$) and female ($n = 19$) C57Bl/6NCrl mice were purchased from Charles River Laboratories. All mice were housed in groups of 3–4 mice per cage and maintained under standard specific pathogen free (SPF) conditions. Mice were given food and water ad libitum in a 12 h light/12 h dark cycle at 21 ± 2 ˚C. Starting at 15 weeks of age, 10 mg/kg isoproterenol ($n = 12$ male, $n = 12$ female) or an equivalent volume of sterile saline ($n = 7$ male, $n = 7$ female) was administered by subcutaneous daily injection for 14 days. Age-matched mice that had been castrated ($n = 4$), ovariectomized ($n = 4$), or sham-operated ($n = 4$ male, $n = 4$ female) by Charles River Laboratories at the age of 4 weeks were subjected to the isoproterenol regimen described above. Mice were monitored for 30 minutes after each isoproterenol injection, and once daily during the prolonged administration to make sure that this dosage regimen is well tolerated. No mortality or significant morbidity were observed in all experimental groups. Animals were humanely euthanized by decapitation under isoflurane anesthesia 1 day after the last injection of isoproterenol. Hearts were collected, rinsed in ice-cold phosphate-buffered saline, snap frozen in liquid nitrogen, and stored at -80˚C.

### Echocardiography

Cardiac function was assessed by echocardiography prior to isoproterenol administration and immediately following the first dose to determine whether there is a sex difference in the inotropic or chronotropic response to acute isoproterenol administration (n = 6 per sex). To determine the response to prolonged isoproterenol administration, cardiac function was assessed by echocardiography following the last dose of isoproterenol or sterile saline injections (n = 7–12 per sex per group). Echocardiography was performed using the Vevo 2100 system (VisualSonics, Inc., Toronto, Ontario, Canada) equipped with an MS400 transducer. Anesthesia was induced with 3% isoflurane in oxygen and maintained at 1–2% during the procedure. Mice were secured in a supine position on a heated physiologic monitoring stage. Parasternal short axis images of the left ventricle were obtained in M-Mode at the level of the papillary muscles. Endocardial and epicardial borders were manually traced over 3–4 cardiac cycles and cardiac output, ejection fraction, fractional shortening, end diastolic and systolic volumes, and left ventricular (LV) Mass were calculated using VisualSonics cardiac measurement package of the Vevo 2100.

### Histopathology

LV heart sections were collected, fixed in 10% neutral buffered formalin and embedded in paraffin. Four-micron sections were stained with hematoxylin and eosin (HE) or Masson's trichrome stain. Histopathologic evaluation was performed by a board-certified veterinary pathologist who was blinded to the experimental group. Each stained HE stained section was examined for (a) inflammation (distribution, severity, and cell type), (b) vascular pathology, (c) interstitial fibrosis, and (d) myofiber degeneration/vacuolization. Inflammation and fibrosis were assessed as follows: 0, absent; 1, minimal inflammation or fibrosis; 2, mild inflammation or fibrosis; 3, moderate inflammation or fibrosis; and 4, marked inflammation or fibrosis. For vascular and myofiber pathology, sections were scored based upon the severity of the change (minimal, mild, moderate, or severe) and the morphologic nature of the pathology. The severity of fibrosis on the trichrome stained section was assessed for fibrosis as described above. Sections from each heart were also immunohistochemically stained for expression of

MAC-2 (galectin-3). In brief, four-micron sections were dewaxed and rehydrated prior to antigen retrieval. Thereafter, sections were incubated with either anti-galectin-3 antibody (clone M3/38, Cedarlane Labs, Burlington, NC) according to manufacturer's instruction. The number of MAC-2 positive cells was manually quantified on the five most cellular 200X images.

## RNA extraction

Total RNA was extracted from 20 mg frozen heart tissue using 300 μL Trizol reagent (Life Technologies, Carlsbad, CA) according to manufacturer's instructions. RNA concentrations were attained by measuring absorbance at 260 nm using a NanoDrop 8000 spectrophotometer (Thermo Fisher Scientific, Wilmington, DE) and first-strand cDNA was synthesized from 1.5 μg total RNA using the high-capacity cDNA reverse transcription kit (Applied Biosystems, Foster City, CA) according to manufacturer's instructions.

## Real-time PCR

Specific mRNA expression was quantified by SYBR Green (Applied Biosystems) based real-time PCR performed on an ABI 7900HT instrument (Applied Biosystems) using 384-well optical reaction plates. Thermocycler conditions were as follows: 95°C for 10 min, followed by 40 PCR cycles of denaturation at 95°C for 15 sec, and annealing/extension at 60°C for 1 min. Gene expression was determined using previously published primers for *ANP*, *BNP*, *and TGF-beta1*. Primer sequences are listed in S1 Table. The mRNA expression levels were normalized to *β-actin* and are expressed relative to male control. Relative gene expression was determined by the ΔΔCT method. Primer specificity and purity of the final PCR product were verified by melting curve analysis.

## Statistical analysis

Data were analyzed using GraphPad Prism software (version 7.01, La Jolla, CA) and are presented as means ± standard errors of the mean (SEM). Comparisons among different sex or surgical alterations and treatment groups were performed by ordinary two-way analysis of variance (ANOVA), followed by Tukey's multiple comparison post-hoc analysis or two-way ANOVA (repeated measures), followed by Sidak's post-hoc analysis where appropriate. Statistical analysis for histopathologic grading and MAC-2 staining was performed using the non-parametric Kruskal-Wallis test. A *p* value of $< 0.05$ was taken to indicate statistical significance.

# Results

## Acute effects of isoproterenol administration on echocardiographic parameters in intact male and female mice

Cardiac function was assessed by echocardiography prior to and immediately following a single subcutaneous injection of 10 mg/kg isoproterenol in adult male and female mice. Representative echocardiographic images obtained in M-Mode from each group are displayed in Fig 1A. Acute administration of isoproterenol had a similar effect on cardiac function in males and females. Ejection fraction and fractional shortening were significantly increased to a similar degree in both sexes. Ejection fraction was increased by 53% in males and 59% in females (Fig 1B) and a 110% increase in fractional shortening was evident in both sexes (Fig 1C). End-systolic volume was decreased by 90% in both sexes (Fig 1D) and end-diastolic volumes were down 35% in male and 30% in female mice (Fig 1E). Cardiac output was significantly increased (35%) in female mice and 20% in male mice, though this increase did not reach statistical significance in males (Fig 1F). As expected, acute treatment with isoproterenol induced

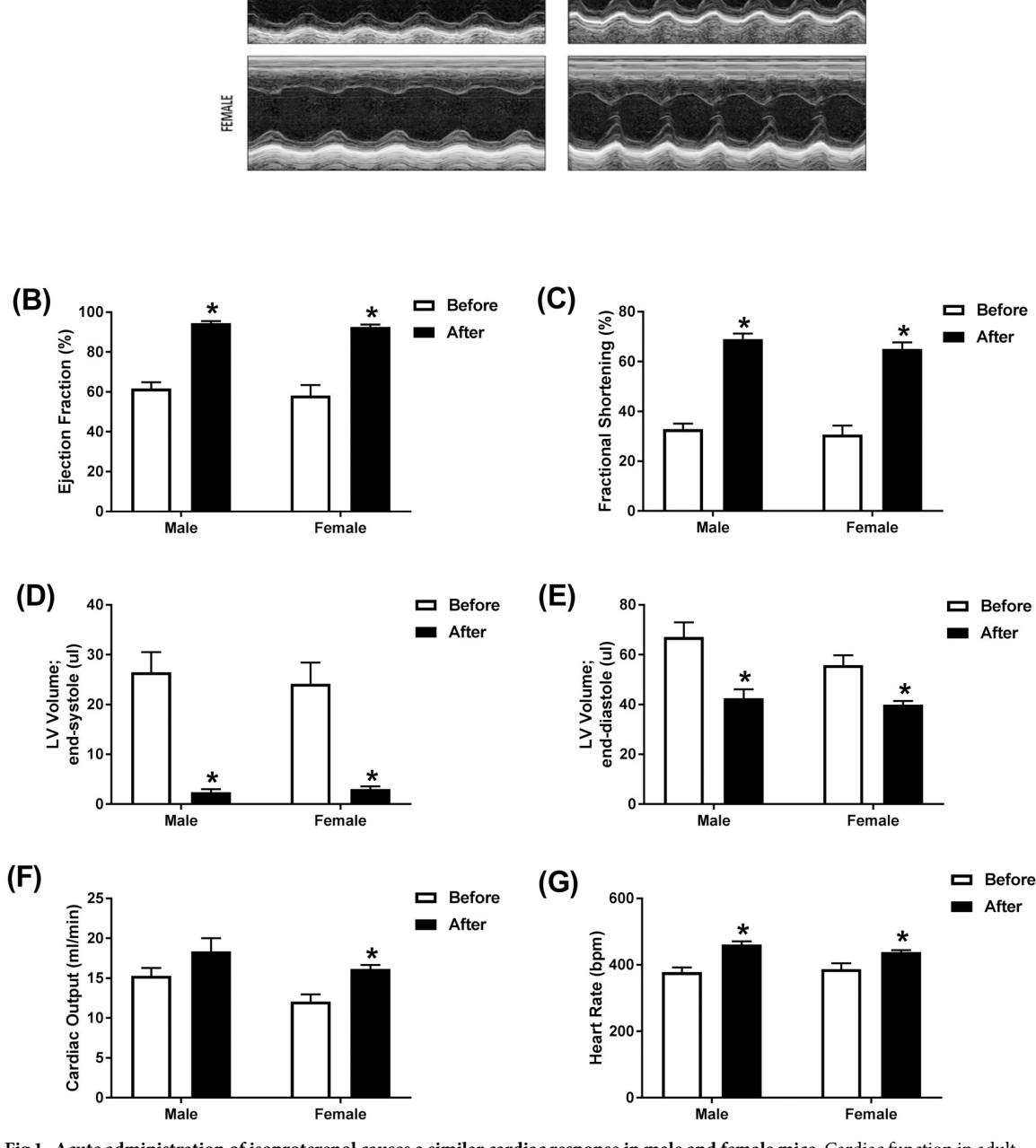

**Fig 1. Acute administration of isoproterenol causes a similar cardiac response in male and female mice.** Cardiac function in adult male ($n = 6$) and female ($n = 6$) C57Bl/6NCrl mice was assessed by trans-thoracic echocardiography just prior to and immediately following a single subcutaneous injection of 10 mg/kg isoproterenol (ISO). (A) Representative images from parasternal short axis view of the heart acquired in M-Mode. Effects of ISO on (B) ejection fraction, (C) fractional shortening, (D) LV volume in end-systole, (E) LV volume in end-diastole, (F) cardiac output, and (G) heart rate. Values are shown as means ± SEM. *$p < 0.05$, compared to before treatment of the same sex by matched two-way ANOVA with Sidak's post-hoc analysis.

a significant increase in heart rate by 22% in male and 13% in female mice (Fig 1G). Matched two-way ANOVA revealed a significant effect of isoproterenol on all the measured parameters, while sex effect was significant only on the cardiac output. There was no significant interaction between isoproterenol and sex in all the measured parameters (S2 Table).

## Chronic administration of isoproterenol caused significant cardiac dysfunction and hypertrophy in intact male and female mice

Cardiac function in male and female mice was assessed by echocardiography following 14 days of treatment with isoproterenol or saline (control). Representative echocardiographic images obtained in M-Mode from each group are displayed in Fig 2A. Chronic administration of isoproterenol had a similar effect on cardiac function in male and female mice. Ejection fraction, an accurate measure of systolic dysfunction [21], and fractional shortening were significantly decreased to a similar degree in both sexes when compared to saline treatment of the same sex. Ejection fraction was decreased by 25% in male and 21% in female mice (Fig 2B). Fractional shortening was decreased by 30% in males and 24% in females (Fig 2C). End-systolic and diastolic volumes were significantly increased in male (87% and 20%, respectively), but not female mice (Fig 2D and 2E). Although not significant, a 61% increase in end-systolic volume was evident in female mice, (Fig 2D) with little change in end-diastolic volume (Fig 2E). Cardiac output was decreased by 22% in male mice and 19% in female mice, though this decrease did not reach significance in female mice (Fig 2F). Female mice had significantly smaller hearts than male mice from the same treatment group (Fig 2G–2H). Isoproterenol-treated mice exhibited a significantly higher heart weight/tibia length ratio (HW/TL) compared to saline-treated mice (Fig 2G). Two-way ANOVA revealed a significant isoproterenol effect on all the measured parameters, while sex effect was significant only in LV Mass and the heart weight/tibia length ratio (HW/TL). There was no significant interaction between isoproterenol and sex in any of the measured parameters (S3 Table).

## Chronic administration of isoproterenol caused significant pathological lesions in the hearts of intact male and female mice

Analysis of the HE- and trichrome-stained heart sections revealed increased fibrosis in both isoproterenol-treated male and female mice (Figs 3 and 4A). The severity of fibrosis did not significantly differ between the male and female mice. Similarly, both male and female treated mice demonstrated significantly higher numbers of MAC-2 positive cells than their control counterparts but were not significantly different from each other (Figs 3 and 4B).

## Chronic administration of isoproterenol induced the gene expression of markers of cardiac remodeling and fibrosis in intact male and female mice

The mRNA expression levels of the natriuretic peptides ANP and BNP were increased, though not significantly, in hearts of mice treated with isoproterenol for 14 days (Fig 5A and 5B). Similar changes were observed in male and female mice in both ANP (2.5- and 3-fold, respectively) (Fig 5A) and BNP (3- and 3.8-fold, respectively) (Fig 5B). Gene expression of the fibrotic marker TGF-beta1 was significantly induced following isoproterenol treatment. A significant 2-fold increase in the gene expression of TGF-beta1 was observed in both male and female mice (Fig 5C). Prolonged isoproterenol administration for 14 days had a significant effect on all the measured genes, while neither sex effect nor the interaction between sex and isoproterenol were statistically significant (S4 Table).

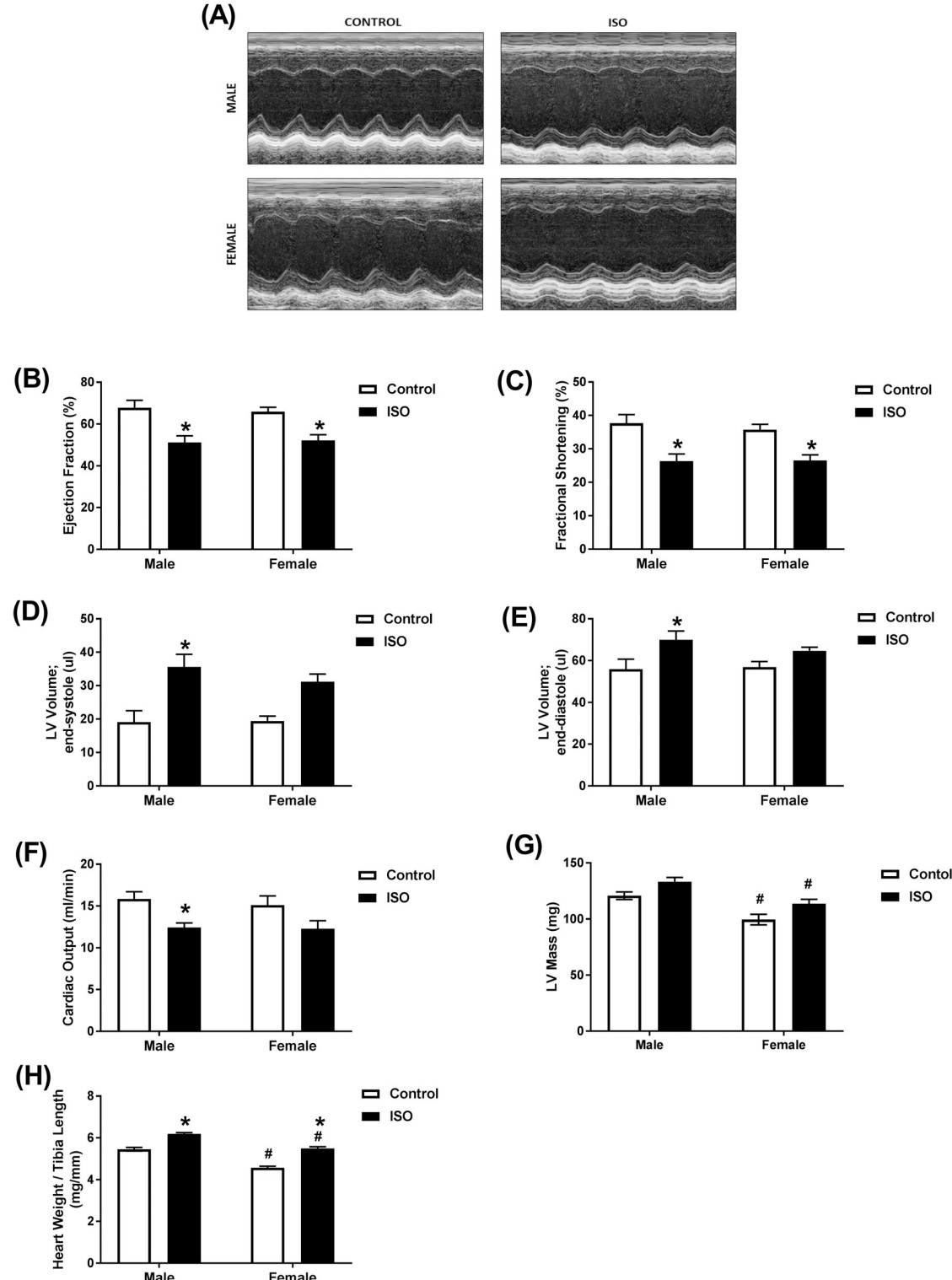

**Fig 2. Prolonged administration of isoproterenol causes similar cardiac dysfunction in male and female mice.** Male and female C57Bl/6NCrl mice were subcutaneously injected with 10 mg/kg isoproterenol (ISO) (male, female *n* = 12) or saline (male, female *n* = 7) daily for 14 days. (A-F) Cardiac function was assessed by trans-thoracic echocardiography. (A) Representative images from parasternal short axis view of the heart acquired in M-Mode. Effects of ISO on (B) ejection fraction, (C) fractional shortening, (D) LV volume in end-systole, (E) LV volume in end-diastole, (F) cardiac output, and (G) LV mass. (H) Heart weight to tibial length ratio. Values are shown as means ± SEM. *$p<0.05$, compared to control treatment of the same sex; # $p<0.05$, compared to male of same treatment by two-way ANOVA with Tukey's post-hoc analysis.

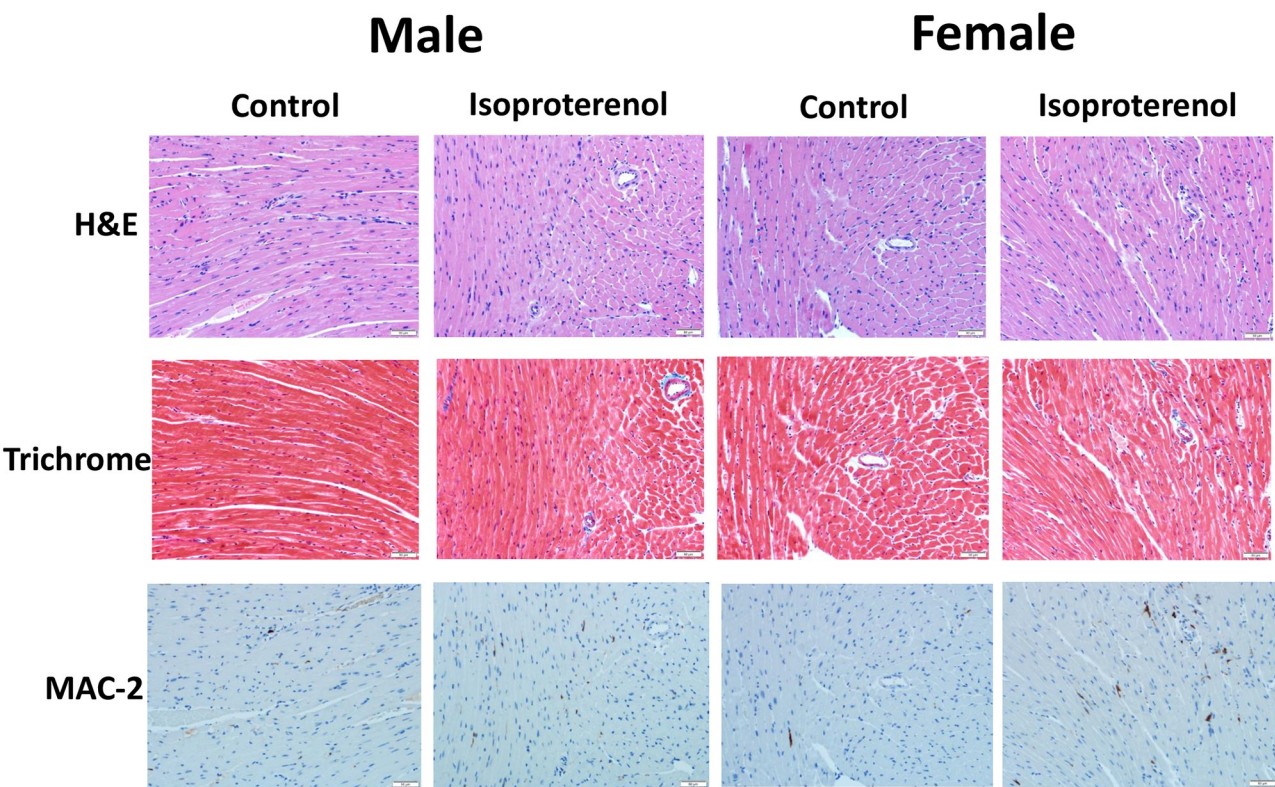

**Fig 3. Prolonged administration of isoproterenol causes similar histopathologic changes in hearts of male and female mice.** Male and female C57Bl/6NCrl mice were subcutaneously injected with 10 mg/kg isoproterenol (ISO) (male, female $n$ = 6, 5, respectively) or saline (male, female $n$ = 5) daily for 14 days. Fig 3 shows representative images from hematoxylin and eosin (HE), Masson's trichrome, and MAC-2 stained sections.

### Effect of castration of male mice on isoproterenol-induced cardiac dysfunction

Isoproterenol treatment for 14 days resulted in similar functional changes in sham and castrated mice. Representative echocardiographic images obtained in M-Mode from each group are displayed in Fig 6A. Ejection fraction (Fig 6B) and fractional shortening (Fig 6C) were significantly reduced by approximately 20% in both sham and castrated mice. Significant increases in end-systolic volumes (Fig 6D) were clearly evident in sham and castrated mice (35 and 56%, respectively). End-diastolic volumes (Fig 6E) were also increased in sham and castrated mice (9 and 15%, respectively), though the increase in sham mice was not statistically significant. Similar decreases in cardiac output (Fig 6F) were observed in castrated and sham mice, though this decrease was not significant in sham animals. Isoproterenol administration caused a significant increase in LV mass in both sham and castrated mice (Fig 6G). Matched two-way ANOVA revealed a significant isoproterenol effect on all the measured parameters. However, the castration effect and the interaction between isoproterenol and castration were not statistically significant (S5 Table).

### Effect of ovariectomy of female mice on isoproterenol-induced cardiac dysfunction

Isoproterenol treatment for 14 days resulted in similar functional changes in sham and ovariectomized mice when compared to measures obtained prior to treatment. Representative

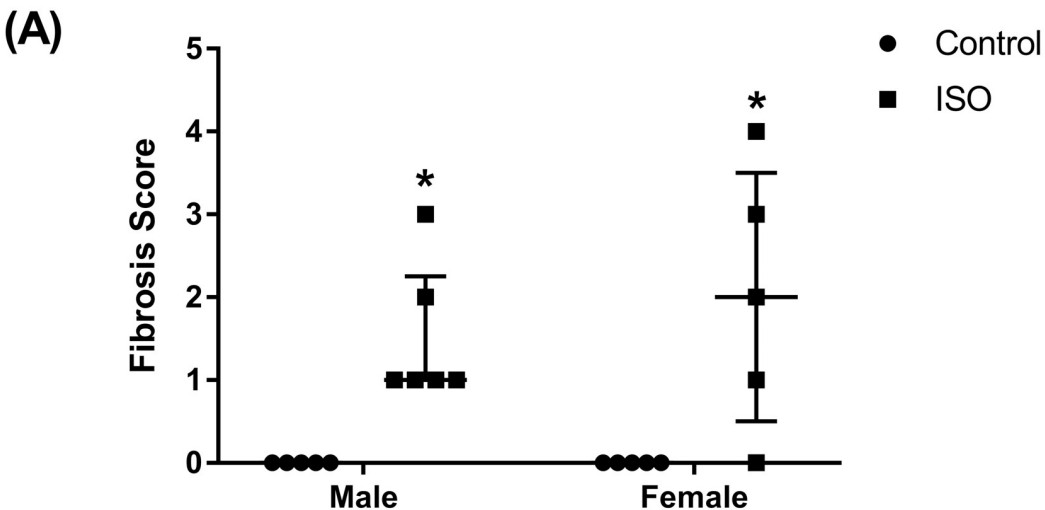

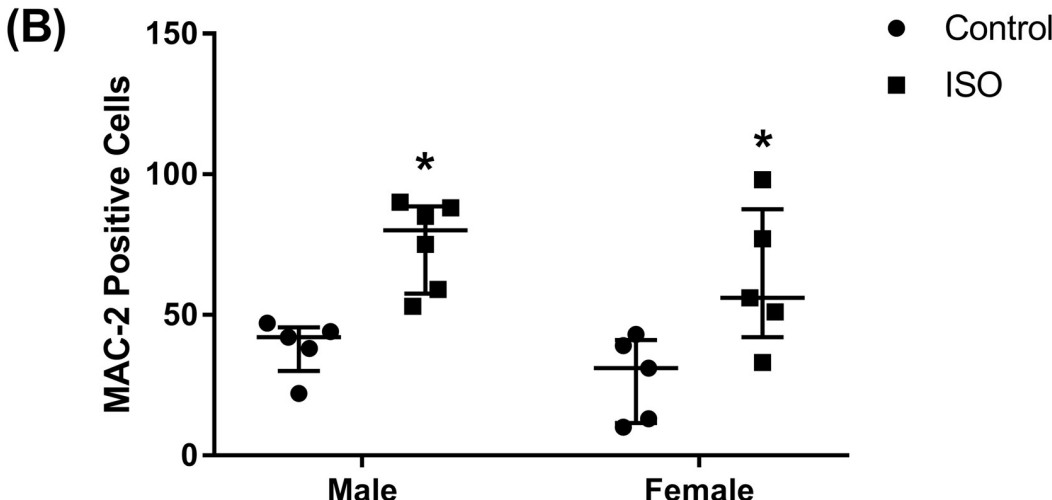

**Fig 4. Prolonged administration of isoproterenol causes similar levels of fibrosis and MAC-2 accumulation in hearts of male and female mice.** Male and female C57Bl/6NCrl mice were subcutaneously injected with 10 mg/kg isoproterenol (ISO) (male, female n = 6, 5, respectively) or saline (male, female n = 5) daily for 14 days. (A) Isoproterenol treated male and female C57Bl/6NCrl mice demonstrate increased collagen (light blue, Masson's trichrome) deposition and (B) increased numbers of MAC-2 positive cells compared to control mice. *$p<0.05$, compared to control treatment of the same sex, by non-parametric Kruskal-Wallis test.

echocardiographic images obtained in M-Mode from each group are displayed in Fig 7A. Ejection fraction and fractional shortening were significantly decreased following isoproterenol treatment in sham (23% and 28%, respectively) and ovariectomized (35% and 42% respectively) mice (Fig 7B and 7C). Isoproterenol treatment resulted in significantly greater end-

**(A)**

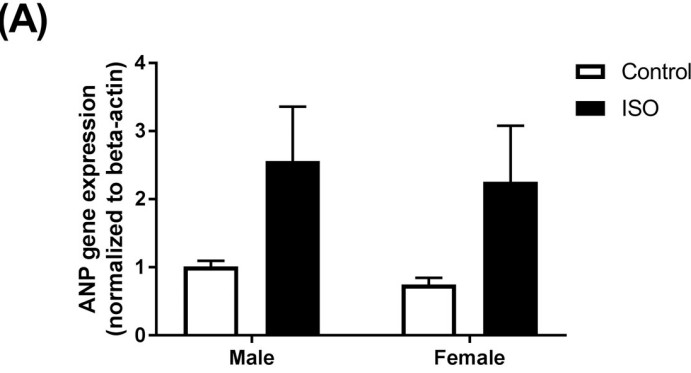

**(B)**

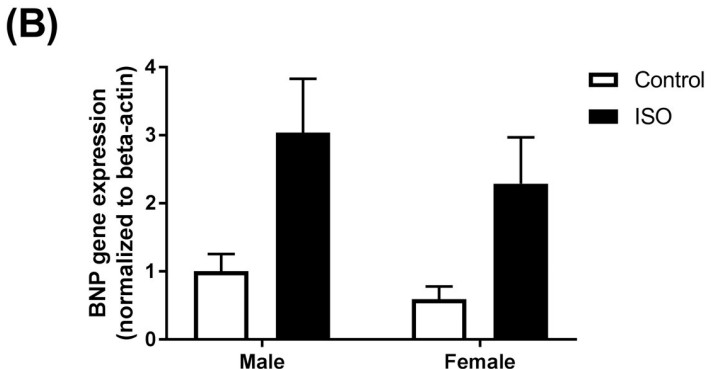

**(C)**

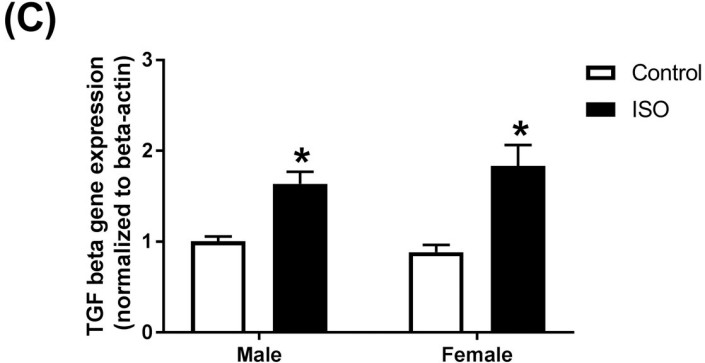

**Fig 5. Prolonged administration of isoproterenol causes similar changes in gene expression of the natriuretic peptides and fibrotic markers in hearts of male and female mice.** Male and female C57Bl/6NCrl mice were given subcutaneous injections of 10 mg/kg isoproterenol (ISO) or saline (control) daily for 14 days ($n$ = 5–6 per group). mRNA expression of (A) ANP, (B) BNP, (C) TGF-beta1 mouse hearts. Results are normalized to beta-actin and expressed relative to male control. Values are shown as means ± SEM. *p<0.05, compared to control treatment of the same sex.

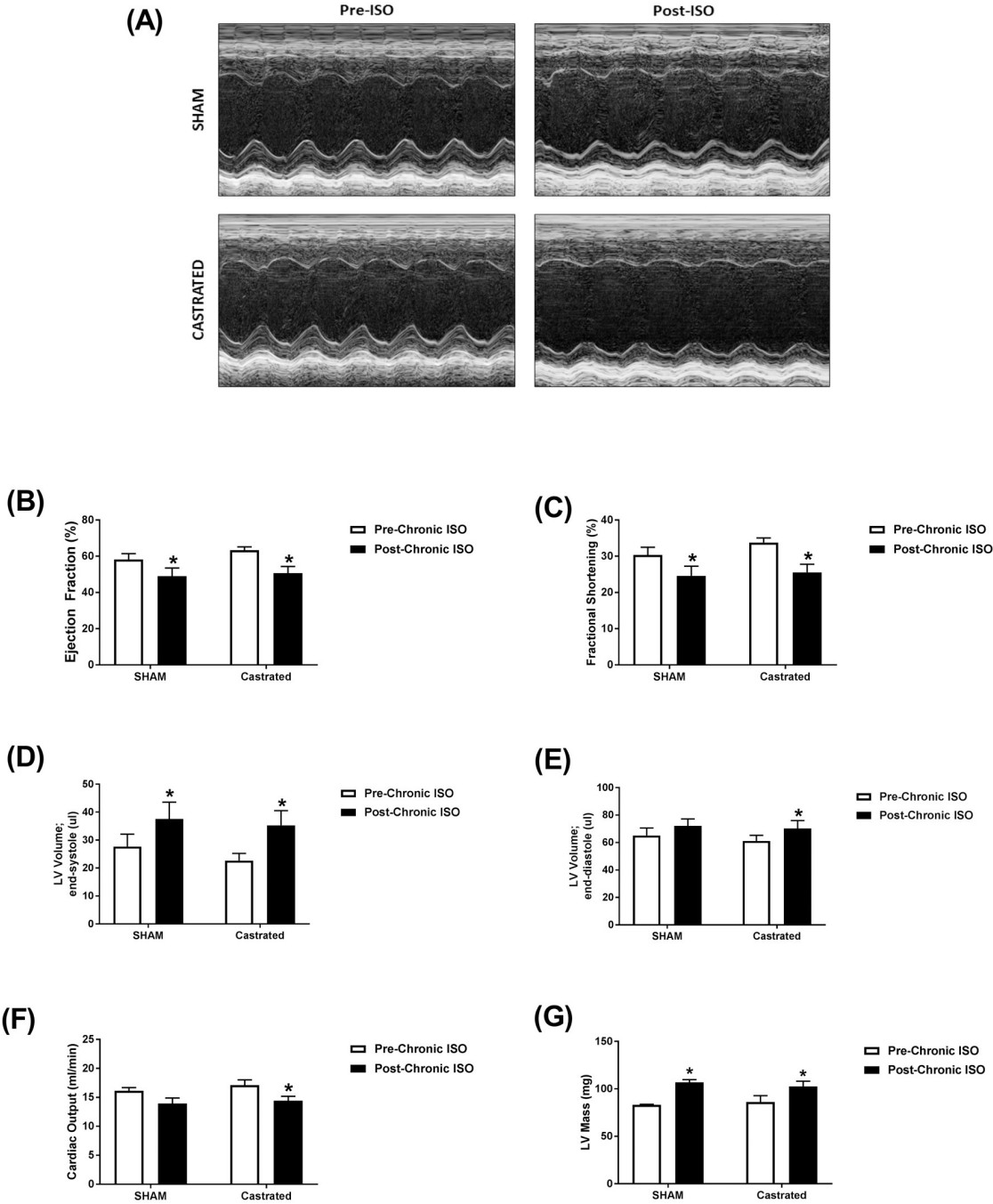

**Fig 6. Prolonged administration of isoproterenol causes similar cardiac dysfunction in castrated or sham-operated (SHAM) male mice.** (A-F) Cardiac function in adult male castrated ($n = 4$) and sham-operated ($n = 4$) C57Bl/6NCrl mice was assessed by trans-thoracic echocardiography one week prior to treatment (Pre-ISO), and again after 2 weeks of daily subcutaneous injections of 10 mg/kg isoproterenol (Post-ISO). (A) Representative images from parasternal short axis view of the heart acquired in M-Mode. Effects of ISO on (B) ejection fraction, (C) fractional shortening, (D) LV volume in end-systole, (E) LV volume in end-diastole, (F) cardiac output, and (G) LV mass. $*p < 0.05$, compared to before treatment of the same surgical status by matched two-way ANOVA with Sidak's post-hoc analysis.

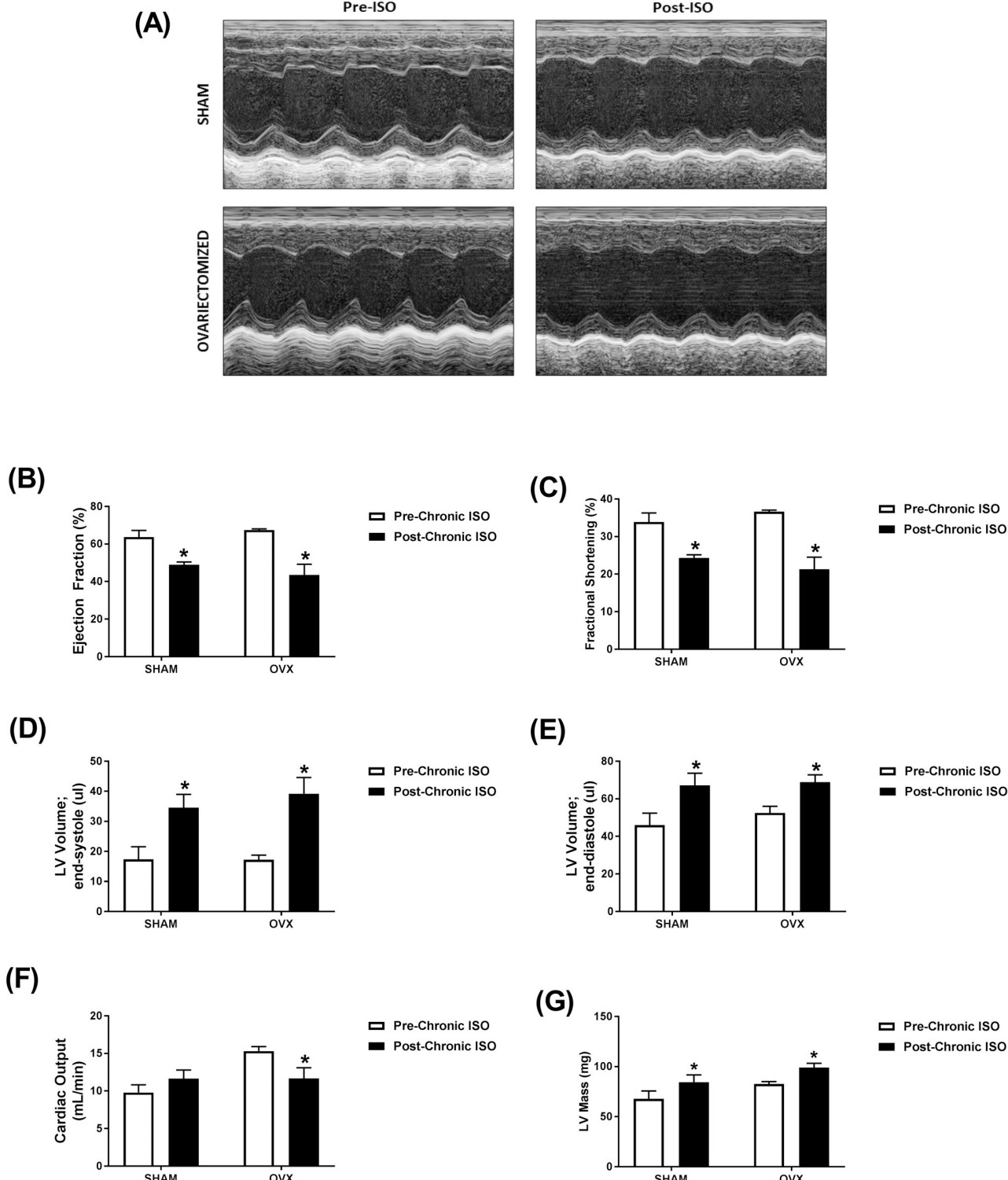

**Fig 7. Prolonged administration of isoproterenol causes similar cardiac dysfunction in ovariectomized (OVX) or sham-operated (SHAM) female mice.** (A-F) Cardiac function in adult female ovariectomized (OVX) ($n$ = 3) and sham-operated ($n$ = 4) C57Bl/6NCrl mice was assessed by trans-thoracic echocardiography one week prior to treatment (Pre-ISO), and again after 2 weeks of daily subcutaneous injections of 10 mg/kg isoproterenol (Post-ISO). (A) Representative images from parasternal short axis view of the heart acquired in M-Mode. Effects of ISO on (B) ejection fraction, (C) fractional shortening, (D) LV volume in end-systole, (E) LV volume in end-diastole, (F) cardiac output, and (G) LV mass. *$p$<0.05, compared to before treatment of the same surgical status by matched two-way ANOVA with Sidak's post-hoc analysis.

systolic and diastolic volumes in both sham (99% and 46%, respectively) and ovariectomized (128% and 31%, respectively) mice (Fig 7D and 7E). Cardiac output was unaltered by isoproterenol treatment in the sham group but was significantly reduced by 20% in ovariectomized mice (Fig 7F). Isoproterenol administration caused a significant increase in LV mass in both sham and ovariectomized mice (Fig 7G). Matched two-way ANOVA revealed a significant isoproterenol effect on all the measured parameters except the cardiac output. However, the ovariectomy effect and the interaction between isoproterenol and ovariectomy were not statistically significant except the cardiac output, where there was a significant interaction (S6 Table).

## Discussion

Sex-related differences have been described in several animal models of cardiac hypertrophy/dysfunction. In models of doxorubicin-induced cardiotoxicity, we and others have demonstrated a profound sexual dimorphism with female rodents being much less sensitive to doxorubicin than male rodents [22–24]. Similarly, female mice were protected against iron overload-induced cardiomyopathy [25]. On the other hand, both male and female rats developed cardiac hypertrophy at 4 and 16 weeks of chronic volume overload. Interestingly, the increase in cardiac muscle mass was greater in females than males at 16 weeks. At 4 weeks of chronic volume overload, however, a decrease in fractional shortening occurred in males only [26]. Pressure overload has been shown to increase left ventricular mass to the same extent in male and female rodents, but with a more marked decline in cardiac function in males [27, 28]. These studies clearly demonstrate that sex-related differences are model-specific.

Isoproterenol-induced cardiac dysfunction is an established model of excessive catecholamine-induced cardiovascular disease [29–31]. While low doses of isoproterenol (0.04 mg/kg/day for 6 months) cause cardiac hypertrophy and fibrosis without significant cardiac dysfunction [17, 18], moderate doses (4–60 mg/kg/day) induce cardiac dysfunction and heart failure [29, 32–34] and high doses (150–300 mg/kg/day) induce a model of myocardial infarction [33, 35]. We selected a moderate dose of isoproterenol (10 mg/kg/day) in our current study because sex-related differences in the response to the low dose have been previously determined [17, 18] and we expected the high dose to cause severe injury that would mask any potential sex-related difference. Therefore, in the current work, we investigated key sex-related differences in the acute response to a moderate isoproterenol dose as well as the response to prolonged administration of isoproterenol (10 mg/kg/day) for 2 weeks to C57Bl/6NCrl mice. After acute administration of isoproterenol, we found out that there was a similar increase in cardiac function parameters such as ejection fraction and fractional shortening as well as a similar increase in heart rates in both male and female mice, indicating similar inotropic and chronotropic effects of isoproterenol in male and female mice. In agreement with these findings, isoproterenol induced similar acute hemodynamic changes in male and female CD-1 mice [20]. Isolated adult rat hearts showed similar chronotropic and inotropic responses to isoproterenol [36]. Male sex was associated with blunted β-adrenergic receptor responsiveness to isoproterenol injection in human volunteers [37]. Contrariwise, cardiomyocytes isolated from male rats had an augmented beta-adrenergic response to isoproterenol [38]. These discrepancies may have arisen due to differences between the *in vivo* and *in vitro* models employed in these studies.

We have also determined the response to prolonged isoproterenol administration in male and female mice. In agreement with human studies [39, 40], the LV mass, the heart weight, and absolute cardiac output values were lower in female than in male mice. However, our data demonstrate that prolonged isoproterenol administration caused a similar extent of cardiac hypertrophy and cardiac dysfunction in both male and female mice. Continuous isoproterenol

administration by mini-osmotic pumps (30 mg/kg/day) to C57Bl/6J mice for one week caused cardiac hypertrophy in both male and female mice; however, females developed significantly less hypertrophy than males [41]. Importantly, this regimen increased fractional shortening in males, but not in female mice [41]. Chronic administration of low isoproterenol doses (0.04 mg/kg/day for 6 months) from 9 to 15 months of age induced cardiac hypertrophy and dilation in male Spontaneously hypertensive rats (SHRs), but not in females [18]. At the molecular levels, we showed that prolonged isoproterenol administration induced the gene expression of the natriuretic peptides, ANP and BNP, to a similar extent in hearts of male and female mice.

In addition to cardiac dysfunction and hypertrophy, male and female C57Bl/6NCrl mice developed a similar degree of myocardial fibrosis in response to chronic isoproterenol administration. There are discrepant reports about the sex-dependent effect of chronic isoproterenol administration on cardiac fibrosis. On one hand, female CD-1 mice showed higher fibrosis than males after chronic isoproterenol administration 7.5 mg/kg/day for 3 weeks [20]. On the other hand, chronic isoproterenol administration 0.04 mg/kg/day for 6 months increased interstitial collagen in male SHRs, but not in females [18]. This discrepancy suggests that sex-related differences in the response to isoproterenol may be dose-dependent, with higher isoproterenol doses causing more severe damage that is not sexually dimorphic. It may also be attributed to strain difference, since male SHRs have a higher extent of baseline collagen content and cardiac hypertrophy than females [42]. In agreement with the similar effect on myocardial fibrosis, we also demonstrated that isoproterenol administration induced the gene expression of several fibrotic markers in hearts of male and female mice to a similar degree. We have also shown that MAC-2 positive cells were equally elevated in the hearts of male and female mice. MAC-2 expression has been shown to be highly associated with myocardial fibrosis [43, 44], although its causative role in inducing myocardial fibrosis is still controversial [45]. Although we did not observe significant differences in these key echocardiographic, histopathologic, and molecular markers of cardiac dysfunction, there may be sex-related differences in the molecular pathways leading to such outcome. This notion is supported by a recent study revealing sex-specific differences in cardiac proteomic and metabolomic profiles between men and women with a similar degree of human heart failure [46] and thus warrants further investigation.

In order to identify the potential role of male sex hormones, we determined the response to prolonged isoproterenol administration in castrated male mice compared to sham-operated animals. Intriguingly, there was no significant difference in the isoproterenol-induced cardiac dysfunction between castrated and sham-operated mice. In agreement with our results, castration had no effect on LV dilation produced by chronic isoproterenol administration 0.015 mg/kg/day for 6 months in Sprague Dawley rats [47]. In contrast, castration prevented cardiac hypertrophy, LV dilation, and increased interstitial collagen produced by chronic isoproterenol administration 0.04 mg/kg/day for 6 months male SHRs [17]. These effects may be attributed to the protective effect of castration on the underlying cardiovascular pathology in male SHR, as previously reported [48], rather than the response to isoproterenol.

In order to identify the role of female sex hormones, we studied the effects of isoproterenol administration in ovariectomized female mice compared to sham-operated controls. Similar to castration in male mice, ovariectomy had no significant effect on isoproterenol-induced cardiac dysfunction. Similar to our findings, ovariectomy did not change the response to chronic isoproterenol administration in female SHRs [17]. Contrariwise, ovariectomized rats developed more extensive cardiac remodeling than intact females in response to chronic volume overload [49]. Estrogen exerted concentration-dependent pro- and anti-hypertrophic response in cultured adult cardiomyocytes isolated from male and female rats [50]. Low estrogen concentrations 1 pM had a pro-hypertrophic effect, whereas high concentration 1 nM prevented

phenylephrine-induced hypertrophy [50]. Estrogen has also been shown to exert a model-dependent effect. Estrogen prevented cardiac hypertrophy and dysfunction caused by pressure overload, but not by myocardial infarction [51]. Although the results of these gonadectomy experiments corroborated our previous findings in intact animals, the small sample size and the lack of saline-treated control mice are limitations to the current work. Therefore, future studies are planned to specifically determine the effects of gonadectomy on isoproterenol-induced cardiac dysfunction and remodeling and to identify the molecular determinants of these effects.

## Conclusion

The current study demonstrates lack of significant sex-related differences in cardiac hypertrophy, dysfunction, and fibrosis in response to moderate-dose isoproterenol (10 mg/kg/day for 14 days) in C57Bl/6NCrl mice. Our findings also demonstrate that gonadectomy of male and female mice did not have a significant impact on isoproterenol-induced cardiac dysfunction. This study suggests that female sex may not be sufficient to protect the heart in this model of isoproterenol-induced cardiac dysfunction. It also signifies the notion that sexual dimorphism in cardiovascular diseases is complex and highly model-dependent.

## Supporting information

**S1 Table. Primer sequences used in this study.**
(DOCX)

**S2 Table. Two-way ANOVA (Repeated Measures) analysis for echocardiographic parameters after acute isoproterenol administration to male and female C57Bl/6NCrl mice.**
(DOCX)

**S3 Table. Two-way ANOVA analysis for echocardiographic parameters after prolonged isoproterenol administration to male and female C57Bl/6NCrl mice.**
(DOCX)

**S4 Table. Two-way ANOVA analysis for hypertrophic and fibrotic markers expression after *prolonged* isoproterenol administration to male and female C57Bl/6NCrl mice.**
(DOCX)

**S5 Table. Two-way ANOVA (Repeated Measures) analysis for echocardiographic parameters after prolonged isoproterenol administration to castrated and sham-operated male C57Bl/6NCrl mice.**
(DOCX)

**S6 Table. Two-way ANOVA (Repeated Measures) analysis for echocardiographic parameters after prolonged isoproterenol administration to ovariectomized and sham-operated female C57Bl/6NCrl mice.**
(DOCX)

## Acknowledgments

Experiments using the NanoDrop 8000, ABI 7900 HT, and Agilent 2100 Bioanalyzer were done with staff support at the University of Minnesota Genomics Center. Experiments using the Vevo 2100 echocardiography system were done with staff support at the University of Minnesota Imaging Center. Processing of heart tissues for histopathological analysis was

performed with staff support at the Comparative Pathology Shared Resource, University of Minnesota Masonic Cancer Center.

## Author Contributions

**Conceptualization:** Beshay N. Zordoky.

**Data curation:** Marianne K. O. Grant, Ibrahim Y. Abdelgawad, Christine A. Lewis, Davis Seelig.

**Formal analysis:** Marianne K. O. Grant, Ibrahim Y. Abdelgawad, Davis Seelig, Beshay N. Zordoky.

**Funding acquisition:** Beshay N. Zordoky.

**Methodology:** Marianne K. O. Grant, Davis Seelig, Beshay N. Zordoky.

**Project administration:** Beshay N. Zordoky.

**Resources:** Beshay N. Zordoky.

**Writing – original draft:** Marianne K. O. Grant, Davis Seelig, Beshay N. Zordoky.

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
