## [Decision Letter · Decision Letter 0]

1 May 2020

PONE-D-20-10894

Isoproterenol-induced Cardiac Dysfunction in Male and Female C57Bl/6 Mice

PLOS ONE

Dear Dr. Zordoky,

Thank you for submitting your manuscript to PLOS ONE. After careful consideration, we feel that it has merit but does not fully meet PLOS ONE’s publication criteria as it currently stands. Therefore, we invite you to submit a revised version of the manuscript that addresses the points raised during the review process.

ACADEMIC EDITOR: All issues raised by reviewers are required.

We would appreciate receiving your revised manuscript by Jun 15 2020 11:59PM. To enhance the reproducibility of your results, we recommend that if applicable you deposit your laboratory protocols in protocols.io, where a protocol can be assigned its own identifier (DOI) such that it can be cited independently in the future. For instructions see: http://journals.plos.org/plosone/s/submission-guidelines#loc-laboratory-protocols

We look forward to receiving your revised manuscript.

Kind regards,

Vincenzo Lionetti, M.D., PhD

Academic Editor

PLOS ONE

Journal Requirements:

1. In your Methods, please provide full details of animal care and housing, including details of the monitoring of the animals for adverse clinical signs.

Reviewers' comments:

Reviewer's Responses to Questions

**Comments to the Author**

1. Is the manuscript technically sound, and do the data support the conclusions?

Reviewer #1: Yes

Reviewer #2: Yes

2. Has the statistical analysis been performed appropriately and rigorously? 

Reviewer #1: I Don't Know

Reviewer #2: I Don't Know

3. Have the authors made all data underlying the findings in their manuscript fully available?

Reviewer #1: Yes

Reviewer #2: Yes

4. Is the manuscript presented in an intelligible fashion and written in standard English?

Reviewer #1: Yes

Reviewer #2: Yes

5. Review Comments to the Author

Reviewer #1: Intention of the study was to explore sex differences in heart function and selected myocardial parameters in the response to acute and prolonged isoproterenol stimulation in C57Bl/6 mice. Findings of the study provide clear message for the readers. However, there are some weak points that should be addressed, in particular those dealing with model and interpretation of the results. Therefore, comments should be taken in mind and revision should be done prior consideration for publication.

Title: In general, title should be informative pointing out the main findings of the study.

Abstract: 14 days of isoproterenol administration cannot be considered as chronic adrenergic stimulation but perhaps prolonged stimulation. Age of the animals should be included when looking for sex differences.

Short Title: This title is not adequate because the study did not show any sex difference.

Key words: Instead sex differences “male and female mice” are more relevant key words.

Introduction: There is no doubts that there are sex-related and age-dependent differences in incidence of CVD and heart failure in human population as shown by national registry based information as well as by numerous studies. The issue is, however, very complex and cannot be simplify by selecting some examples without explanations. Therefore, it would be worthwhile to know why there were no sex-related differences in studies you have cited. Which factors appears to play a role in addition to sex hormones.

You noted that sexual dimorphism in cardiovascular diseases is complex and context-dependent. What does mean context-dependent?

Having proper background, you can hypothesize and test your hypothesis. What should imitate your mice model? What was the reason for selected dosing of isoproterenol and duration of application?

Methods: Do you think that a single dose 10 mg/kg of isoproterenol may induce cardiac dysfunction in mice? Maybe higher dose would be more appropriate when examining sex differences. Moreover, as you know rodents are quite resistant to drugs mostly due to high metabolic rate (related also to high heart rate comparing to other species) Some studies did use higher dose do examine extent of ISO-induced injury, e.g. Wallner 2017.

Do you think that mice model is relevant to imitate clinical condition when taking into consideration that there were no differences between males and females controls in your registered parameters?

Results: It is appreciated using thoracic echo examination to detect heart function in mice. How about accuracy of heart function changes in such small animals?

Discussion should be revised to use more proper explanation of the findings and to note limitations of the study.

Conclusions: The findings of the current study can hardly be translating to humans. Moreover, the study did not use severe pathologic stimulus (according to changes in histology, heart function, etc). Therefore, following sentence is not adequate. This study suggests that female sex may not be sufficient to protect the heart against a severe pathologic stimulus, which may explain why women are susceptible to Takotsubo cardiomyopathy.

Reviewer #2: Overview and general recommendation:

Sex differences in cardiovascular disease has been observed in clinical medicine, as well as within various animal models of cardiac disease. The exact manifestation of these differences varies considerably between species and models. This study investigated the sex differences in a model of excessive catecholamine-induced cardiovascular disease disease by administration of moderate (10 mg/kg) dose isoproterenol subcutaneously to male and female mice. Additionally, the role of sex hormones in this model was further investigated by use of gonadectomized male and female mice. Echocardiographic parameters, histopathology of myocardial tissue, and gene expression of fibrotic markers were assessed.

Overall, this paper is well-written and clear, and the conclusions drawn appear appropriate based on the data provided. The authors discuss the relation between their findings and a multitude of other published studies investigating the role of sex in models of cardiovascular disease in laboratory rodents. The authors are honest about the shortcomings of the second arm of the study regarding gonadectomized animals and do not seek to draw substantial conclusions from this data.

2.1 BROAD COMMENTS

Methods: It would be helpful to clarify the amount of time that has passed from gonadectomy to use in this study for the altered mice. This will alleviate any concerns regarding an incomplete washout period, where residual sex hormones may still be circulating.

Discussion: It is my opinion that a brief discussion of the variables that were of significant difference between sexes (CO in the acute phase, LV mass and HW/TL ratio in the chronic phase), and why these differences still allow the author to draw the conclusion that this model lacks significant sex-related differences, would strengthen the authors position. These items are currently left out of the discussion, and may leave readers wondering if these factors are being intentionally ignored.

2.2 SPECIFIC COMMENTS

Line 63: It would be helpful to characterize the class of pharmacologic agent.

Line 64: Recommend clarifying that this is used to induce cardiac pathologies in laboratory animal models.

Line 81: Recommend providing the exact strain name at least in the Methods section (C57BL/6NCrl)

Line 106: Specify the type of trichrome stain

Line 210: Recommend adding “animals” after the last “sham” of this sentence.

Line 221: Either replace “ISO” with isoproterenol or introduce this abbreviation earlier

Line 246-248: The wording of this sentence is such that it seems to say that you have found sex-related differences, when your conclusion is in fact the opposite. Instead of determined, consider “investigated”, or similar terminology to clarify that you were looking in to the possibility of these differences, not that they were determined.

Figure 6: Suggest matching the language between the legend (CAST) and figure itself (CASTRATED)

6. PLOS authors have the option to publish the peer review history of their article (what does this mean?). If published, this will include your full peer review and any attached files.

Reviewer #1: Yes: Narcisa Tribulova

Reviewer #2: Yes: Amanda L Carlson

---

## [Author Response · Author response to Decision Letter 0]

8 Jun 2020

Response to Reviewers:

We would like to thank the editor and reviewers for their time and effort in reviewing our first submission and for providing their valuable comments which have improved the quality and clarity of the manuscript. Please find the following detailed response to the raised concerns.

Editor’s Comments:

1. In your Methods, please provide full details of animal care and housing, including details of the monitoring of the animals for adverse clinical signs.

Response: We would like to thank the editor for this recommendation. We have updated the Methods section in the revised manuscript to include the requested details. Please find lines 94 105 in the revised manuscript.

“Intact male (n=19) and female (n=19) C57Bl/6NCrl mice were purchased from Charles River Laboratories (Protocol ID: 1807-36187A). All mice were housed in groups of 3 - 4 mice per cage and maintained under standard specific pathogen free (SPF) conditions. Mice were given food and water ad libitum in a 12 h light/12 h dark cycle at 21 ± 2 °C. Starting at 15 weeks of age, 10 mg/kg isoproterenol (n=12 male, n=12 female) or an equivalent volume of sterile saline (n=7 male, n=7 female) was administered by subcutaneous daily injection for 14 days. Age-matched mice that had been castrated (n=4), ovariectomized (n=4), or sham-operated (n=4 male, n=4 female) by Charles River Laboratories at the age of 4 weeks were subjected to the isoproterenol regimen described above. Mice were monitored for 30 minutes after each isoproterenol injection, and once daily during the prolonged administration to make sure that this dosage regimen is well tolerated. No mortality or significant morbidity were observed in all experimental groups”.

Response: The captions for the Supporting Information have been added to the revised manuscript. Please find lines 632 – 646.

 

Reviewer #1: Intention of the study was to explore sex differences in heart function and selected myocardial parameters in the response to acute and prolonged isoproterenol stimulation in C57Bl/6 mice. Findings of the study provide clear message for the readers. However, there are some weak points that should be addressed, in particular those dealing with model and interpretation of the results. Therefore, comments should be taken in mind and revision should be done prior consideration for publication.

Title: In general, title should be informative pointing out the main findings of the study.

Response: We would like to thank the reviewer for this suggestion. We agree with the reviewer about the importance of an informative title. The title of the revised manuscript has been changed to a more informative one. The title now reads: Lack of Sexual Dimorphism in a Mouse Model of Isoproterenol-induced Cardiac Dysfunction.

Abstract: 14 days of isoproterenol administration cannot be considered as chronic adrenergic stimulation but perhaps prolonged stimulation. 

Response: We agree with the reviewer that the word “prolonged” better describes our isoproterenol dosage regimen. The word “chronic” has been changed to “prolonged” in the abstract (line 21) and all over the manuscript (please refer to the marked copy of the revised manuscript).

Age of the animals should be included when looking for sex differences.

Response: We would like to thank the reviewer for this important suggestion. The age of mice is now mentioned in the abstract (line 24).

Short Title: This title is not adequate because the study did not show any sex difference.

Response: We agree with the reviewer about this point. The short title has been changed to “Lack of Sex Differences in Cardiac Dysfunction”.

Key words: Instead sex differences “male and female mice” are more relevant key words.

Response: We agree with the reviewer about this pint. The key words have been changed to: Sex; male and female; Isoproterenol; Cardiac Dysfunction; Cardiac Hypertrophy

Introduction: There is no doubts that there are sex-related and age-dependent differences in incidence of CVD and heart failure in human population as shown by national registry based information as well as by numerous studies. The issue is, however, very complex and cannot be simplify by selecting some examples without explanations. Therefore, it would be worthwhile to know why there were no sex-related differences in studies you have cited. Which factors appears to play a role in addition to sex hormones.

Response: We would like to thank the reviewer for giving us the opportunity to expand on the introduction section of this manuscript. In the revised version, we have re-written the introduction to elaborate more on the referenced studies. Since the sex-related differences in cardiovascular diseases are complex and cannot be adequately covered in the limited space of the Introduction section, we have also cited important references that provide detailed discussions in this area. 

Please refer to lines 55 – 60

For instance, despite similar overall risk of heart failure between men and women, heart failure with reduced ejection fraction (HFrEF) is more prevalent in men, while heart failure with preserved ejection fraction (HFpEF) is more prevalent in women [6, 7]. Although the rationale for the higher prevalence of these specific conditions in women is still not fully understood, coronary microvascular dysfunction and endothelial inflammation have been suggested to play key roles [6].

And lines 64 – 74

Although this traditional view has offered the biological underpinning for several cardiovascular pathologies that are more prevalent and more severe in males [8, 9], it could not explain other clinical and preclinical observations including: the worse outcome of post-menopausal women receiving supplemental estrogen therapy which was mainly attributed to increased rate of thromboembolic events [10], in addition to post-ischemic cardioprotection in aromatase knock-out mice [11] and worsening of heart failure in castrated male experimental animals [12, 13] which suggest a cardioprotective role of androgens. In addition to sex hormones, recent studies suggest that inherent differences in sex chromosome genes may contribute to sex differences in cardiovascular diseases [14].These studies, among others, strongly suggest that sexual dimorphism in cardiovascular diseases is highly complex in humans (reviewed in [6, 8, 14-16]) and thus warrants further investigations in a variety of models in laboratory animals.

You noted that sexual dimorphism in cardiovascular diseases is complex and context-dependent. What does mean context-dependent?

Response: We agree with the reviewer that the word “context-dependent” is vague, so we have changed it in the revised manuscript to “model-dependent”.

Please find lines 19 – 20 in the abstract: Sex-related differences in cardiovascular diseases are highly complex in humans and model-dependent in experimental laboratory animals.

Having proper background, you can hypothesize and test your hypothesis. What should imitate your mice model? What was the reason for selected dosing of isoproterenol and duration of application?

Response: We would like to thank the reviewer for these suggestions. Previous sex differences studies used lower isoproterenol doses that did not cause cardiac dysfunction. Therefore, we initiated this project as an exploratory project, not hypothesis driven. According to the reviewer’s suggestion, we explained the rationale behind the selected isoproterenol dosage regimen in the Discussion section of the revised manuscript.

Please refer to lines 272 – 275

We selected a moderate dose of isoproterenol (10 mg/kg/day) in our current study because sex-related differences in the response to the low dose have been previously determined [17, 18] and we expected the high dose to cause severe injury that would mask any potential sex-related difference.

Methods: Do you think that a single dose 10 mg/kg of isoproterenol may induce cardiac dysfunction in mice? Maybe higher dose would be more appropriate when examining sex differences. Moreover, as you know rodents are quite resistant to drugs mostly due to high metabolic rate (related also to high heart rate comparing to other species) Some studies did use higher dose do examine extent of ISO-induced injury, e.g. Wallner 2017.

Response: We would like to thank the reviewer for raising up this point. The rationale for the acute isoproterenol administration was not clear in the first submission. We are aware that a single injection of isoproterenol at 10 mg/kg will not cause immediate cardiac dysfunction. Our intention was to determine whether there is a sex difference in the positive inotropic and chronotropic effects of isoproterenol. This rationale is better explained in the revised manuscript.

Please refer to lines 110 – 113 in the Methods section:

Cardiac function was assessed by echocardiography prior to isoproterenol administration and immediately following the first dose to determine whether there is a sex difference in the inotropic or chronotropic response to acute isoproterenol administration (n=6 per sex).

And lines 277 – 281 in the Discussion section:

After acute administration of isoproterenol, we found out that there was a similar increase in cardiac function parameters such as ejection fraction and fractional shortening as well as a similar increase in heart rates in both male and female mice, indicating similar inotropic and chronotropic effects of isoproterenol in male and female mice.

Do you think that mice model is relevant to imitate clinical condition when taking into consideration that there were no differences between males and females controls in your registered parameters?

Response: We would like to thank the reviewer for this comment. Indeed, we found differences in baseline cardiac morphometry between male and female mice, in agreement with human studies. However, there was no significant difference in the response to isoproterenol. These baseline differences are now better articulated in the revision.

Please refer to lines 200 – 201 in the Results

Female mice had significantly smaller hearts than male mice from the same treatment group (Fig. 2G-H).

And lines 290 – 291 in the Discussion

In agreement with human studies [39, 40], the LV mass, the heart weight, and absolute cardiac output values were lower in female than in male mice.

Results: It is appreciated using thoracic echo examination to detect heart function in mice. How about accuracy of heart function changes in such small animals?

Response: We recognize that trans-thoracic echocardiography in small animals is not an easy job. In this study, echocardiography has been performed by the first author who is well trained on small animal echocardiography and has performed more than 400 echocardiography measurements on mice. This method has accurately measured the changes in LV function in mice showing an increase in EF upon acute adrenergic stimulation a reduction in EF upon prolonged administration. The results of LV mass are corroborated by the heart weight data obtained after necropsy. 

Discussion should be revised to use more proper explanation of the findings and to note limitations of the study.

Response: We would like to thank the reviewer for this comment. The Discussion has been updated in the revised manuscript to discuss study limitations. 

Please refer to lines 318 – 323

Although we did not observe significant differences in these key echocardiographic, histopathologic, and molecular markers of cardiac dysfunction, there may be sex-related differences in the molecular pathways leading to such outcome. This notion is supported by a recent study revealing sex-specific differences in cardiac proteomic and metabolomic profiles between men and women with a similar degree of human heart failure [46] and thus warrants further investigation.

And lines 345 – 350

Although the results of these gonadectomy experiments corroborated our previous findings in intact animals, the small sample size and the lack of saline-treated control mice are limitations to the current work. Therefore, future studies are planned to specifically determine the effects of gonadectomy on isoproterenol-induced cardiac dysfunction and remodeling and to identify the molecular determinants of these effects.

Conclusions: The findings of the current study can hardly be translating to humans. Moreover, the study did not use severe pathologic stimulus (according to changes in histology, heart function, etc). Therefore, following sentence is not adequate. This study suggests that female sex may not be sufficient to protect the heart against a severe pathologic stimulus, which may explain why women are susceptible to Takotsubo cardiomyopathy.

Response: We agree with the reviewer that these sentences in the Conclusion may not be the best to reflect our findings. Therefore, we have changed these sentences in the Conclusion to read: “This study suggests that female sex may not be sufficient to protect the heart in this model of isoproterenol-induced cardiac dysfunction. It also signifies the notion that sexual dimorphism in cardiovascular diseases is complex and highly model-dependent” (lines 356 -358).

Regarding the translatability of the findings, experimental laboratory animals have been extensively used to study sex-related differences in cardiovascular diseases. Although there is no perfect animal model that can faithfully mirror a specific human disease, using a variety of models will provide valuable information about the biological basis of these diseases.

Reviewer #2: Overview and general recommendation:

Sex differences in cardiovascular disease has been observed in clinical medicine, as well as within various animal models of cardiac disease. The exact manifestation of these differences varies considerably between species and models. This study investigated the sex differences in a model of excessive catecholamine-induced cardiovascular disease disease by administration of moderate (10 mg/kg) dose isoproterenol subcutaneously to male and female mice. Additionally, the role of sex hormones in this model was further investigated by use of gonadectomized male and female mice. Echocardiographic parameters, histopathology of myocardial tissue, and gene expression of fibrotic markers were assessed.

Overall, this paper is well-written and clear, and the conclusions drawn appear appropriate based on the data provided. The authors discuss the relation between their findings and a multitude of other published studies investigating the role of sex in models of cardiovascular disease in laboratory rodents. The authors are honest about the shortcomings of the second arm of the study regarding gonadectomized animals and do not seek to draw substantial conclusions from this data.

2.1 BROAD COMMENTS

Methods: It would be helpful to clarify the amount of time that has passed from gonadectomy to use in this study for the altered mice. This will alleviate any concerns regarding an incomplete washout period, where residual sex hormones may still be circulating.

Response: We would like to thank the reviewer for this important comment. Gonadectomy was performed by Charles River Laboratories at the age of 4 weeks to ensure adequate washout period of sex hormones. This information has been added to the revised manuscript (lines 101 – 102).

Discussion: It is my opinion that a brief discussion of the variables that were of significant difference between sexes (CO in the acute phase, LV mass and HW/TL ratio in the chronic phase), and why these differences still allow the author to draw the conclusion that this model lacks significant sex-related differences, would strengthen the authors position. These items are currently left out of the discussion, and may leave readers wondering if these factors are being intentionally ignored.

Response: We would like to thank the reviewer for this excellent suggestion. We have discussed these parameters in the revised version.

Please refer to lines 290 – 291

In agreement with human studies [39, 40], the LV mass, the heart weight, and absolute cardiac output values were lower in female than in male mice.

2.2 SPECIFIC COMMENTS

Line 63: It would be helpful to characterize the class of pharmacologic agent.

Response: Lines 75 – 76 now read: “Isoproterenol is non-selective beta-adrenergic agonist commonly used as a pharmacological agent to induce a spectrum of cardiac pathologies in laboratory animal models”.

Line 64: Recommend clarifying that this is used to induce cardiac pathologies in laboratory animal models.

Response: Lines 75 – 76 now read: “Isoproterenol is non-selective beta-adrenergic agonist commonly used as a pharmacological agent to induce a spectrum of cardiac pathologies in laboratory animal models”.

Line 81: Recommend providing the exact strain name at least in the Methods section (C57BL/6NCrl)

Response: We would like to thank the reviewer for this excellent suggestion, since there are important differences between the C57Bl/6J and the C57Bl/6NCrl strains of mice. In the revised submission, we have changed all C57Bl/6 mentions to C57Bl/6NCrl. Please refer to the marked copy of the revised submission.

Line 106: Specify the type of trichrome stain

Response: We specified that we used Masson’s trichrome stain. Please refer to the Histopathology section in the revised submission.

Line 210: Recommend adding “animals” after the last “sham” of this sentence.

Response: This is corrected in the revised submission (line 234 in the revised submission)

Line 221: Either replace “ISO” with isoproterenol or introduce this abbreviation earlier

Response: This is corrected in the revised submission (line 245 in the revised submission)

Line 246-248: The wording of this sentence is such that it seems to say that you have found sex-related differences, when your conclusion is in fact the opposite. Instead of determined, consider “investigated”, or similar terminology to clarify that you were looking in to the possibility of these differences, not that they were determined.

Response: We would like to thank the reviewer for this excellent comment. In the revised submission, we have changed “determined” to “investigated” all over the manuscript. Please refer to the marked copy of the revised submission (multiple occurrences).

Figure 6: Suggest matching the language between the legend (CAST) and figure itself (CASTRATED)

Response: Corrected with thanks.

---

## [Decision Letter · Decision Letter 1]

23 Jun 2020

PONE-D-20-10894R1

Lack of Sexual Dimorphism in a Mouse Model of Isoproterenol-induced Cardiac Dysfunction

PLOS ONE

Dear Dr. Zordoky,

Thank you for submitting your manuscript to PLOS ONE. After careful consideration, we feel that it has merit but does not fully meet PLOS ONE’s publication criteria as it currently stands. Therefore, we invite you to submit a revised version of the manuscript that addresses the points raised during the review process.

ACADEMIC EDITOR: minor issues are required

We look forward to receiving your revised manuscript.

Kind regards,

Vincenzo Lionetti, M.D., PhD

Academic Editor

PLOS ONE

Reviewers' comments:

Reviewer's Responses to Questions

**Comments to the Author**

1. If the authors have adequately addressed your comments raised in a previous round of review and you feel that this manuscript is now acceptable for publication, you may indicate that here to bypass the “Comments to the Author” section, enter your conflict of interest statement in the “Confidential to Editor” section, and submit your "Accept" recommendation.

Reviewer #1: All comments have been addressed

Reviewer #2: (No Response)

2. Is the manuscript technically sound, and do the data support the conclusions?

Reviewer #1: Yes

Reviewer #2: Yes

3. Has the statistical analysis been performed appropriately and rigorously? 

Reviewer #1: Yes

Reviewer #2: I Don't Know

4. Have the authors made all data underlying the findings in their manuscript fully available?

Reviewer #1: Yes

Reviewer #2: Yes

5. Is the manuscript presented in an intelligible fashion and written in standard English?

Reviewer #1: Yes

Reviewer #2: Yes

6. Review Comments to the Author

Reviewer #1: Thanks for your valuable responses that for sure improved the understanding and message of your study.

Reviewer #2: Ms. Ref. No.: PONE-D-20-10894

Title: Isoproterenol-induced Cardiac Dysfunction in Male and Female C57Bl/6 Mice

Grant, Abdelgawad, Lewis, Seelig, and Zordoky

Overview and general recommendation:

The authors have thoroughly addressed this reviewer’s concerns, and the paper is well written and draws valid conclusions.

SPECIFIC COMMENTS

Line 131 – 133: The scale for histopathology currently reads, “0, absent; 1, minimal inflammation or fibrosis; 2, mild minimal inflammation or fibrosis; 3, moderate minimal inflammation or fibrosis; and 4, marked minimal inflammation or fibrosis.”. Is the word “minimal” in 2, 3, and 4 an error?

7. PLOS authors have the option to publish the peer review history of their article (what does this mean?). If published, this will include your full peer review and any attached files.

Reviewer #1: No

Reviewer #2: No

---

## [Author Response · Author response to Decision Letter 1]

23 Jun 2020

We would like to thank the editor and reviewers for their time and effort in reviewing our first submission and for providing their valuable comments which have improved the quality and clarity of the manuscript. Please find the following detailed response to the raised concerns.

Reviewer #1: Thanks for your valuable responses that for sure improved the understanding and message of your study.

Response: We are truly appreciative to the reviewer for their comments.

Reviewer #2: 

Overview and general recommendation:

The authors have thoroughly addressed this reviewer’s concerns, and the paper is well written and draws valid conclusions.

Response: We are truly appreciative to the reviewer for their comments.

SPECIFIC COMMENTS

Line 131 – 133: The scale for histopathology currently reads, “0, absent; 1, minimal inflammation or fibrosis; 2, mild minimal inflammation or fibrosis; 3, moderate minimal inflammation or fibrosis; and 4, marked minimal inflammation or fibrosis.”. Is the word “minimal” in 2, 3, and 4 an error?

Response: We would like to thank the reviewer for catching this typographical error. The word “minimal” in 2, 3, and 4 was an error and omitted in the revised submission.

Lines 130 – 133 now read: “Inflammation and fibrosis were assessed as follows: 0, absent; 1, minimal inflammation or fibrosis; 2, mild inflammation or fibrosis; 3, moderate inflammation or fibrosis; and 4, marked inflammation or fibrosis”.

---

## [Decision Letter · Decision Letter 2]

25 Jun 2020

Lack of Sexual Dimorphism in a Mouse Model of Isoproterenol-induced Cardiac Dysfunction

PONE-D-20-10894R2

Dear Dr. Zordoky,

We’re pleased to inform you that your manuscript has been judged scientifically suitable for publication and will be formally accepted for publication once it meets all outstanding technical requirements.

Kind regards,

Vincenzo Lionetti, M.D., PhD

Academic Editor

PLOS ONE

Additional Editor Comments (optional):

Reviewers' comments:

Reviewer's Responses to Questions

**Comments to the Author**

1. If the authors have adequately addressed your comments raised in a previous round of review and you feel that this manuscript is now acceptable for publication, you may indicate that here to bypass the “Comments to the Author” section, enter your conflict of interest statement in the “Confidential to Editor” section, and submit your "Accept" recommendation.

Reviewer #2: All comments have been addressed

2. Is the manuscript technically sound, and do the data support the conclusions?

Reviewer #2: Yes

3. Has the statistical analysis been performed appropriately and rigorously? 

Reviewer #2: I Don't Know

4. Have the authors made all data underlying the findings in their manuscript fully available?

Reviewer #2: Yes

5. Is the manuscript presented in an intelligible fashion and written in standard English?

Reviewer #2: Yes

6. Review Comments to the Author

Reviewer #2: (No Response)

7. PLOS authors have the option to publish the peer review history of their article (what does this mean?). If published, this will include your full peer review and any attached files.

Reviewer #2: No

---

## [Editor Report · Acceptance letter]

29 Jun 2020

PONE-D-20-10894R2 

Lack of Sexual Dimorphism in a Mouse Model of Isoproterenol-induced Cardiac Dysfunction 

Dear Dr. Zordoky:

I'm pleased to inform you that your manuscript has been deemed suitable for publication in PLOS ONE. Congratulations! Your manuscript is now with our production department. 

Kind regards, 

on behalf of

Prof. Vincenzo Lionetti 

Academic Editor

PLOS ONE